# Barriers and facilitators to addressing mental health needs among Asian, Black and Latin American men who have sex with men (MSM) in England and Wales: A qualitative study

Andrew Ghobrial[1]*, Phil Samba[1,2], Fiona M. Burns[1], Emily Jay Nicholls[1], Peter Weatherburn[3], Fiona C. Lampe[1], Isaac Yen-Hao Chu[1,3], Alison J. Rodger[1], T. Charles Witzel[1]

1 Institute for Global Health, University College London, London, United Kingdom, 2 The Love Tank CIC, London, United Kingdom, 3 Department of Public Health, Environments and Society, London School of Hygiene and Tropical Medicine, London, United Kingdom

* rmhajgh@ucl.ac.uk

## Abstract

Ethnic and sexual minority groups are underserved by mental health services globally despite having potentially greater need. This study aimed to explore how the intersections between sexual orientation and ethnicity shape mental health experiences and service access for Asian, Black and Latin American men who have sex with men (MSM) in the UK. This research was drawn from a qualitative sub-study of a larger HIV self-testing randomised controlled trial (SELPHI). Cis-gender Black, Asian and Latin American MSM who participated in SELPHI were recruited purposively to ensure sample diversity. Semi-structured interviews including a focused section on mental health were conducted between April and July 2020. A thematic framework approach was used to analyse the transcribed interview data. Twenty-nine participants were interviewed, comprising thirteen Black, eleven Asian and five Latin American MSM. The data were organised into three meta-themes exploring 1) Background, culture and upbringing, 2) Sexuality and manifestation of mental health issues, and 3) Barriers and facilitators to accessing mental health services. Childhood experiences of hypermasculine norms shaped the development of self-reliant coping strategies for mental distress. Peer support was protective of mental health, but alcohol, party drugs and chemsex could exacerbate feelings of isolation. Intersectional stigma restricted mental health service access, highlighting the need for culturally competent services. Previous use of mental health services and openness about mental health among social groups were facilitators to access. Private mental health services were often favoured due to perceptions of a superior quality of care and the speed of access, although participants recognised this as a financial barrier which further deepened structural inequities in access to mental healthcare. This study highlights the importance of multi-system and interdisciplinary interventions to facilitate discussions surrounding mental health within Asian, Black and Latin American MSM communities. In particular, services

**Data availability statement:** Due to their sensitive and personally identifiable nature, underlying data will not be made freely available. Requests will be considered by the LSHTM Research Ethics Committee. Please contact ethics@lshtm.ac.uk.

**Funding:** This manuscript presents independent research funded by the National Institute for Health and Care Research (NIHR) under the Programme Development Grants (Reference number: NIHR203298 to AJR). The views expressed in this manuscript are those of the author(s) and not necessarily those of the NIHR or the Department of Health and Social Care. The funders had no role in study design, data collection and analysis, decision to publish, or preparation of the manuscript.

**Competing interests:** I have read the journal's policy and the authors of this manuscript have the following competing interests: TCW received speaking honoraria from Gilead Sciences between 2017 and 2023. The remaining authors have declared that no competing interests exist.

must be mindful of the barriers and facilitators faced by these groups when accessing mental health services, including norms linking self-reliance and masculinity.

## Introduction

Sexual minority groups have been identified as having more unmet physical and mental health needs than their heterosexual counterparts as a result of multi-level stigma, structural marginalisation and social exclusion [1–6].

Ethnic minority populations also face similar health inequities such as unequal access to resources and services, which are typically not adapted to or designed for priority groups [7–10]. In addition, exposure to racial discrimination has been shown to have negative impacts on physical and mental health outcomes [11,12], and increases the chance of potential risk-taking sexual behaviours including condomless sex and multiple sexual partners [13].

The complex and overlapping experiences of stigma and discrimination based on both ethnicity and sexual orientation have been described, although the literature has been inconsistent in its conclusions with regards to the relationship between mental health, ethnic minority status and being LGBTQ+. There is evidence to support theoretical frameworks such as the minority stress model [14], which elucidates some of the structural elements that predispose minority groups to poorer health and well-being outcomes, in addition to the impact of compounding experiences of stigma and discrimination through having multiple minoritised identities [15–23].

Belonging to a sexual minority within a minoritised ethnic group is associated with higher rates of depression, anxiety and suicidal ideation [24–26], with contributing factors including experiences of racism within LGBTQ+ communities [27,28] as well as heterosexism and possible rejection from their ethnic communities [29–32]. Sexual racism has been identified as a distinct predictor of poor mental health outcomes [33]. Enacted stigma, for example increased rates of police harassment, is associated with increased sexual risk-taking among Black MSM and transgender women [34]. Manifestations of structural racism, homophobia and transphobia, for example poorer educational attainment and financial exclusion among gender and sexual minorities, also specifically correlate with poorer physical and mental health outcomes [35–36]. This was disproportionately amplified among gender and sexual minority Black, Asian and Latinx people after the COVID-19 pandemic [37]. Experiences of stigma continue to contribute towards a disproportionate burden of poor mental health among MSM today [38].

There has been criticism of the minority stress theory in other academic work as this model may overlook the importance of resilience. Indeed, there is a body of evidence to suggest that belonging to a minoritised ethnic group, or participating in LGBTQ+ communities is protective against mental health problems [39–41]. Indeed, resilience appears to be gained from sharing a collective identity within a community or through adaptive strategies built from experiencing discrimination during ones' upbringing [42–44], and through supportive parenting [45]. These protective factors have also been demonstrated among minority ethnic groups where sexual orientation is not accounted for [46–48].

Problematic alcohol and drug use have long been identified as areas for targeted public health interventions for MSM. Worse mental health outcomes do not necessarily correlate with drug use in itself, but are sometimes associated with sexualised drug use, poly drug use and drug dependence [49–50]. Chemsex is primarily characterised by use of one or a combination of mephedrone, gamma-hydroxybutyrate (GHB) and methamphetamine used during sexual encounters to facilitate increased pleasure and longevity [51,52]. Associations between

chemsex harms and the ethnicity of sexual minorities are complex [39,53–56] Although White MSM are more likely to engage in chemsex [57–60] there is some evidence of a greater risk of psychological morbidity and drug dependency with substance use among Latino and Asian American sexual minorities in the United States [29,59,61,62].

The disparities in the literature demonstrate the complexity of factors at play and the importance of considering intersectionality as more nuanced than simply a sum of identities [24,63–66], with consideration for age, migration status, religion, romantic involvement and social cohesion [67–70]. There is also significant variation in how mental health is evaluated with differing proxy outcomes between studies, including standardised or descriptive categorisation of symptoms, mental health diagnoses or hard outcomes such as self-harm or suicide; in addition to variations in how discrimination and stigma are assessed [71].

The discrepancies also illuminate limitations in evaluating significantly heterogenous groups such as 'LGBTQ+ people' or 'minority ethnic groups, ' as sexual and gender minorities face some overlapping but also many different stressors [72]. Indeed, various ethnic groups have nuanced and distinct experiences of prejudice and unique protective factors [27,73,74] LGBTQ+ people also experience distinct differences in their mental health. Bisexual people have a higher rate of mental health concern and underrepresentation compared with gay or lesbian people [75–78]. Sexual minority men are disproportionately affected by depression and anxiety compared with sexual minority women, although women who have sex with women have a higher risk of drug or alcohol dependency [79–84].

Despite the complexities in understanding the nuanced intersections between race, ethnicity, gender and sexual orientation, the literature is consistent in outlining that people identifying as sexual, gender and ethnic minorities have largely unmet heath and social needs, are often unreached by public health interventions and face distinct barriers to accessing support [1,17,75–78,85,86]. There is a pressing public health need for interventions to improve access to and provision of mental healthcare among this group [79–82,84]

It is important to note that the majority of work exploring the mental health of ethnically minoritised LGBTQ+ people has been conducted with cisgender MSM in the United States, with a dearth of evidence from other settings. Asian, Black and Latin American MSM in the UK are vastly underrepresented in health research [77,87], but existing literature demonstrates pronounced physical and mental health inequalities experienced by ethnic minority MSM in the UK. [80,83,88–90].

The migration and cultural histories of these ethnic groups in the UK are distinct from MSM in the United States, where health service provision also differs substantially [91,92]. Currently mental health services in the UK face distinct economic and service-level pressures [93,94]. Although mental health services are free at the point of use as part of the National Health Service (NHS), funding for services is England and Wales is apportioned autonomously by 42 localised Integrated Care Boards (ICBs) in England and seven Local Health Boards in Wales. This leads to considerable disparities in mental health service configuration and provision around the country and regional variations in access to services [95,96]. Accessibility issues have further been compounded by a previous government policy of austerity which led to the closure of some specialist services and capacity issues at many others [97,98]. Primary care services which provide a basic level of mental healthcare and are the most frequent route for accessing specialist mental health services have also been under increasing pressure [93,99]. In addition, there is substantial private sector provision of mental health services (largely psychology services) in the UK, which are paid for either by the individual or through corporate employment schemes [100].

This study aims to explore how the intersections between sexual orientation and ethnicity shape mental health experiences and service access for Asian, Black and Latin American MSM in England and Wales.

We do this by developing an understanding of how culture, family and social networks shape understandings of poor mental health; how mental health challenges are impacted by the LGBTQ+ scene; and exploring barriers and facilitators to mental health service access.

Acknowledging the heterogeneity between the groups included, this study is intended as a starting point for further work to explore more nuanced differences between ethnic, gender and sexual minorities in the UK.

## Materials and methods

This research is part of a series of qualitative sub-studies from the SELPHI (An HIV Self-testing Public Health Intervention) randomised controlled trial, which recruited 10,135 MSM (cis and trans) and trans women to investigate the impact of providing HIV self-testing (HIVST) on HIV diagnosis rates [101–103]. SELPHI embedded qualitative research in its methodology to examine HIVST intervention acceptability, feasibility and to explore the experiences of underrepresented populations in the trial, including Asian, Black and Latin American MSM [78,104]. This qualitative study was designed to explore the unmet mental health needs of these ethnically minoritized MSM and identify service provision considerations that may improve their access to mental healthcare. In-depth interviews were used to develop a nuanced understanding of individual experiences.

### Inclusion criteria

Men were eligible for this study if they were SELPHI participants and of Asian, Black or Latin American ethnicity (using adapted UK standardised ethnicity codes). All participants recruited to this study were in the intervention arms of the SELPHI RCT (e.g., they received one or more HIVST kits through the trial).

### Sampling

We sought a diverse sample with regard to ethnicity, age and educational attainment. Eligible men were contacted by TCW and provided with a participant information sheet. Consenting men were interviewed online (using Zoom) because of COVID-19 restrictions in place at the time, by one of PS, EJN or TCW. We conducted 25 interviews, reviewed the interim demographic data, and conducted a further 4 interviews with Black African MSM as this group was underrepresented in the research. Participants were provided £30 for participation.

### Data generation

The topic guide was adapted from previous SELPHI qualitative sub-studies [102,103,105], with additional questions tailored to this specific population (S1 File: Interview topic guide). Acknowledging this group has specific mental health need needs which are poorly understood, we included a focused section on experiences of, and attitudes towards, mental health challenges including accessing support. This was co-developed with the study peer researcher PS. This explored: conversations individuals had about mental health with friends and family; experiences of poor mental health and service access; alcohol and drug use and preferences for accessing support.

Interviews commenced on 15 April 2020 and finished on 15 July 2020. 28 interviews were conducted by EJN, PS and TCW over video-conferencing software and one using instant

messages due to domestic privacy concerns. Interviews were audio recorded, transcribed verbatim by a third-party service, checked for accuracy and then anonymised.

### Analysis

A thematic framework analysis was undertaken to analyse the data. Sections of the transcripts addressing mental health were selected for analysis by AG after familiarisation with the text. An initial framework was developed for analysis by AG and TCW through developing themes from the data and reviewing the existing literature [106]. This was piloted on two transcripts and revised for clarity with PS. Further revisions to this coding framework were made iteratively as the transcripts were coded and applied retrospectively to previously coded manuscripts. We used QSR NVivo 12 for data organisation. Throughout the analysis AG, TCW and PS met to discuss the and refine the themes as a team, ensuring trustworthiness. Input was especially sought from PS as the participant and public involvement lead for the grant. This ensured the analysis was grounded the in the key issues described by the participants and reflected the needs of the community.

### Ethics and consent to participate

All participants provided verbal recorded consent. Ethical approval was granted by University College London [ref: 9233/001] on 13 March 2020, and the London School of Hygiene and Tropical Medicine [ref: 21837] on 2 April 2020. SELPHI was prospectively registered with the ISRCTN [ref: ISRCTN 20312003]. Further ethical approval for secondary analysis of qualitative data was granted by UCL on 7 March 2023 [ref: 24477.001].

Due to their sensitive and personally identifiable nature, underlying data will not be madefreely available. Requests will be considered by the LSHTM Research Ethics Committee.Please contact ethics@lshtm.ac.uk.

## Results

Twenty-nine cis-gender MSM were interviewed, all of Asian, Black and Latin American ethnicity. Ten participants were aged 18-25 years, ten were aged 26-35, three were aged 36-45 and six participants were older than 46. Fourteen participants were of Black ethnicity, seven of whom were Black Caribbean or Mixed Black Caribbean, six were Black African or Mixed Black African and one was of Other Black ethnicity. Eleven participants were of Asian or Mixed Asian ethnicity, and four participants were Latin American. Ninety-two percent of participants were educated above GCSE level (UK school-leaving qualification at age 16), and 57% had a degree or a higher qualification. Table 1 provides further socio-demographic data for the study participants.

Data were organised into three meta-themes exploring 1) Background, culture and upbringing, 2) Sexuality and manifestation of mental health issues, and 3) Barriers and facilitators to accessing mental health services. These themes also represent sequential perspectives of past, present and future respectively, and describe how experiences of their past influence the participants' current experiences and shape their attitudes towards accessing services in the future. The meta-themes and themes are presented in Table 2.

### Meta-theme 1: Background, culture and upbringing

The first meta-theme explores our participants' backgrounds, highlighting specific cultural factors or aspects of their upbringing that have shaped their perspectives on, and experience of, mental health issues.

Table 1. Participant demographics.

| Demographic | Description | Count |
|---|---|---|
| Age | 18–25 | 10 |
| | 26–35 | 10 |
| | 36–45 | 3 |
| | 46⁺ | 6 |
| Ethnicity | Black/ Black Caribbean incl. mixed | 7 |
| | Black/ Black African incl. mixed | 6 |
| | Black (other) | 1 |
| | Asian Pakistani/ Asian Indian | 4 |
| | Asian Chinese | 2 |
| | Asian (other)/ Mixed Asian | 5 |
| | Latin American | 4 |
| Sexual Orientation | Gay | 23 |
| | Bisexual | 2 |
| | Other/ undisclosed | 4 |
| Highest educational qualification | GCSEs (General Certificate of Secondary Education) and below | 2 |
| | GCE A-levels (General Certificate of Education) or equivalent higher education below degree level | 11 |
| | Degree or higher | 16 |
| Region | East Midlands | 1 |
| | East of England | 1 |
| | London | 19 |
| | North West England | 1 |
| | South East England | 2 |
| | West Midlands | 4 |
| | Yorkshire and the Humber | 1 |
| Last HIV test | <12 months | 19 |
| | >12 months | 7 |
| | Never | 3 |

Table 2. Summary of meta-themes and themes.

| Meta-themes | 1: Background, culture and upbringing | 2: Sexuality and manifestation of mental health issues | 3: Barriers and facilitators to accessing mental health services |
|---|---|---|---|
| Themes | 1.1: Family and acceptance | 2.1: Openness and ability to discuss mental health issues | 3.1: Previous service use as a facilitator to future encounters |
| | 1.2: Childhood trauma, abuse and early sexualisation | 2.2 Embedded parts of gay life | 3.2: Financial means, quality of care and access |
| | | 2.3: Vulnerability and masculinity | 3.3: Stigma and perceptions of mental health issues |
| | | | 3.4: Structural factors, language and geography |

## Theme 1.1: Family and acceptance

This theme describes the influence of openness within family household while growing up and its relationship to the participants' coming to terms with their sexual orientation. Participants

perceived both being gay and experiencing poor mental health as being undesirable characteristics, a belief that was reinforced in households that did not encourage conversations around either or both issues. For some participants, these family tensions persisted through their adult life; one participant conducted the interview via instant messaging for his safety, as he felt it was too dangerous to speak audibly about his sexuality in his family home.

The study participants were from a broad range of backgrounds but many shared common experiences of specific cultural or religious values within family households, with an emphasis on stoicism, resilience, success in education and achieving financial security or good social standing through employment. Such values were often entangled with specific understandings of masculinity, and meeting these expectations frequently came at the cost of presenting a true version of themselves with regards to their sexual orientation. The importance of these values linking stoicism and self-reliance to masculinity was common across all groups, but for men from Black African and Black Caribbean backgrounds ideas around stoicism were especially linked with a social role around masculine behaviours. For men of Asian backgrounds self-sufficiency and success in education/employment were more emphasised. For all participants who described the impact of these values, these conceptions of masculinity required suppression of their internal struggles and of their sexuality, both of which were at odds with the values instilled in them during their upbringing.

> *"I try, but given the Asian background, mental health means visible cues, like you can't talk or do something … you've got a degree, doing a good job, there is no reason for mental health problem. So for them, it's not mental health. It's just like a man being a man."* 26-35, Indian, unreported sexual orientation, London

> *"So having African culture growing up, sexuality was never talked about so it was a secret and the father in the house at the time had a very toxic masculinity atmosphere, he was abusive while also didn't have time for sadness or crying and we had to be 'real men' from young."* 18-25, Mixed White & Black African, gay, London

Conversely, participants who grew up in an environment that encouraged open discussion about mental health felt that this was something they became accustomed to. These men were equipped with a language that allowed them to express themselves more comfortably, and many had developed strategies for managing mental health problems.

> *"But then I just go to the gym or walk the dog or something like that. And I know how to manage my mental health … We would just sit and chat 'cause* [my dad] *was a mental health nurse. And a lot of his friends were mental health nurses. So they'd all be round the house. And they're West Indian too. So you'd be playing dominoes with them and they'd be chatting."* 46+, Mixed White & Black Caribbean, gay, London

### Theme 1.2: Childhood trauma, abuse and early sexualisation

Coming out as gay was traumatic for many participants, who felt that the lack of acceptance by their community shaped their own self-acceptance. This in turn influenced their capacity to manage their mental health and difficulties with self-esteem.

> *"A* lot *of people who aren't straight growing up will have experienced very, very similar types of anxiety about* [coming out] *and I think that is something that inevitably will just bleed into conversations about mental health because I feel like a lot of people have the same kind of background for why they have mental health issues."* 18-25, Mixed White & Asian, bisexual, South East England

Participants who had experienced sexual abuse during childhood described a direct link with profound struggles in adulthood, including depression, anxiety and addiction. This was felt also to be associated with a lack of understanding of what healthy relationships should look like in the wake of sexual abuse, and internalised shame linked to feeling that their sexual orientation was transgressive during a formative time in their lives as a result of said abuse.

*"The local priest took an interest in me and things went awry after that. There was a sexual assault there which started me drinking. My dad died … a few weeks after that situation, and then the drinking started not long after that. So I was only just 18 when that kind of stuff happened."* 26-35, Mixed White & Black Caribbean, unreported sexual orientation, London

Experiences of physical and emotional abuse in relationships exacerbated existing trauma and feelings of disempowerment associated with being outed to unaccepting families. One participant describes his existing anxiety and depression progressed to suicidal ideation at multiple times during his life, some as early as adolescence.

*"I have anxiety,* [generalised anxiety disorder] *and* [seasonal affective disorder], *and also severe depression. I have attempted/contemplated suicide on multiple occasions and most recently when my ex-partner was abusive towards me and also the other time it was most serious was when I was 15-18 with me being outed to my family about my sexuality."* 18-25, Mixed White & Black African, gay, London

## Meta-theme 2: Sexuality and manifestation of mental health issues

This meta-theme moves the analysis to present-day experiences of the participants in recognising and managing their mental health needs. For some, strategies included conversations within supportive social circles, whilst others used more self-reliant coping strategies such as exercise, meditation or creative outlets. For some participants, coping with or compartmentalising mental distress involved self-described excessive use of alcohol, drugs or sex, and some discussed these as deep-seated aspects of gay life were difficult to extract themselves from when they felt vulnerable.

### Theme 2.1: Openness and ability to discuss mental health issues

As a result of a need to conceal and/or suppress their sexuality, some participants described being less open about their mental health in their social circles, and others made reference to developing a parallel life around their sexual orientation that they became adept at hiding from family, friends, colleagues and health professionals. This facilitated a loss of the social structures that provided emotional guardrails against self-destructive practices.

Most participants felt that inclusion within a network of gay people was a protective factor for their mental health, and described how this camaraderie with others, with whom they shared similar experiences, mitigated the lack of understanding from people outside of this community. This understanding from peers helped individuals to cope with emotionally difficult situations and increased resilience in the face of adversity and discrimination.

*"I definitely do think there is something different about talking to someone who is not straight… I guess it's the shared experience of kind of bleakness sometimes… I think it's by nature, being part of any kind of minority group, you have in-jokes about the shared experiences that you have that are not good."* 18-25, Mixed White & Asian, bisexual, South East England

Participants reported that having an awareness of mental health issues was an important part of the process of coming to terms with their sexuality and trying to overcome homophobia. One gay interviewee described how this experience contrasted with that of his bisexual friends:

> *"The gay friends I have would be similar. The bisexual friends I have, I think it's different. Most of my personal sexual relationships are with bisexual men, and I find that their… how they deal with things is very different than my gay male friends*[...] *I do think there's been a lot of work around gay men being able to really express who they are. And so I feel, over the years, I've seen a movement forward. And I feel like bisexual men have been left out of that."* 46+, Black (other), gay, London

Participants felt more comfortable having open conversations with friends and peers who were more aligned with themselves demographically, particularly with regards to ethnicity.

> *"If it's a gay,* white *friend then we may talk about things and our experiences might be slightly different. If it's a gay, black friend, our experiences may be similar in some respects."* 46+, Mixed White & Black Caribbean, unreported sexual orientation, Yorkshire

A small minority of participants however felt that their sexuality had no bearing on discussions about mental health among their friends and made a distinction between problems relating to their sex lives or relationships and problems relating to their mental health. For these men, their sexual orientation appeared more removed from their sense of self.

> *"When you're talking about mental health, unless it's directly related to the act of having sex, I don't see why somebody's sexuality would be relevant."* 36-45, Mixed White & Black African, gay, South East England

Finally, some participants discussed generational differences in openness about mental health problems. Younger participants often commented on having clearer conversations about their mental health with friends, compared to older participants who described more indirect ways of addressing this topic in conversation. This suggests a possible shift in how mental health has been considered and discussed between generations.

> *"It's something that always come up. So when we're talking about someone. I think our generation in general are more open about it so, yes."* 18-25, Latin American, gay, London

### Theme 2.2 Embedded parts of gay life

Alcohol and drug-taking were frequently referred to as a part of the gay party scene that for some of the interviewees felt like an integral part of their social lives. This had an influence on their responses to difficult circumstances, and for some of the participants was a central component of their coping strategies as well as their social and sexual practices.

> *"I've always taken drugs, I always drank, I always had sex… When I was growing up, this was a really, really gay area as well… you had loads of gay pubs. So it's just always there …"* 46+, Mixed White & Black Caribbean, gay, London

Even for participants who did not take drugs, encounters with other men who did, especially during hook-ups, provided an insight into how difficult it could be to distance oneself

from the chemsex scene. For those not engaged in chemsex, sexual encounters with others who were could increase feelings of isolation, loneliness, vulnerability and lead to unfamiliar and unwelcome risks.

*"I went on Grindr [...] And he came to my house and he brought a bottle of drink. And I mixed it with other alcohol. And then it was afterwards that he told me that it was GHB or something [...] I really felt I put myself in a really vulnerable situation. I think I was just vulnerable and so into doing my escort and massage work. I was probably feeling very low and isolated and a bit probably needing some sort of social contact."* 46+, Mixed White & Black Caribbean, gay, London

Some interviewees described a link between their mental health and their sexual activity. For these men, sex could be used as a coping strategy during periods of challenging circumstances. Indeed, adversity was often a catalyst for having more sex and provided a distraction from difficulties.

*"Sexual behaviour, yes, I think on a minor scale, in terms of* [patching] *loneliness - in the same way someone will use drink or drugs. When you don't want to think about something, you would - I would fill it with people."* 46+, Black (other), gay, London

This also appeared to be a source for shame, as one participant explained that periods of poor mental health linked with periods of self-judgement around having multiple sexual partners.

*"It goes up and down depending on my mental health. 'Cause I also have issues around depression. So if I'm feeling depressed, then I'll probably have quite a lot of negative self-talk about being promiscuous."* 46+, Mixed White & Black Caribbean, gay, London

### Theme 2.3: Vulnerability and masculinity

Themes of masculinity and vulnerability emerged during the interviews, and participants with experiences of hypermasculine environments growing up reflected on this when talking about the expectations they held of themselves in the present.

For some individuals, notions of masculinity led to the establishment of defence structures. Resilience to adversity was felt to be a representation of strength and masculinity, and many participants, especially from Asian and Latin American backgrounds, made reference to self-sufficiency when explaining that they did not often have conversations about their struggles with friends and family.

*"I've built up my own defence system. I know and allow more time and then I can understand my problems - and I know how to get on with it."* 26-35, Asian Indian, unreported sexual orientation, London

One participant described how expectations of young Black men conforming to hypermasculine norms led to expectations in relationships to deflect or disguise vulnerability. This could also manifest as a sense of imperviousness to illness and harm. In this particular example, Black men were perceived by the participant to be less likely to enact health-seeking behaviours (including testing for HIV and accessing other health services) and less likely to access mental health services because of social imperatives surrounding 'toughness' that they needed to preserve.

*"I just think of how someone is younger coming into this [...] They're already having to deal with the black Superman complex, which is this [...] I'm sure you know that whole thing of this hyper-masculine, they can't be vulnerable… and then you have the possibility of an intimate situation that allows you to be vulnerable. But then the partners you're with are saying, no, you can't be vulnerable. […] you're constantly trying to be impenetrable. […] Well, the mental health around that, I think, it's holding that up. And then I think you're less likely to test* [for HIV], *simply because, in many ways, you've taken on the assumption that you're invulnerable. […] And then supporting men of colour. 'Cause I don't think it's just Black men that hold onto those things. And how to support them and then they are allowed to deflate. They are allowed to be vulnerable."* 46+, Black (other), gay, London

## Meta-theme 3: Barriers and facilitators to accessing mental health services

This meta-theme discusses the participants' perspectives and experiences of using mental health services, highlighting barriers to accessing care and facilitators to engaging with services. We also explore determinants of their use of mental health services in the future.

### Theme 3.1: Previous service use as a facilitator to future encounters

Twenty of the twenty-nine participants in this study had accessed support for their mental health. Twelve reported accessing specialist NHS mental health services via a referral from primary care. Another two participants were seen only by their GP and managed with medication without an onward mental health team referral. Two further participants had experience of using psychology services through their higher education institutions, two through community-led non-profit organisations and one participant had experience of solely using private mental health services. Two other participants reported accessing private mental healthcare in addition to having used NHS mental health services previously.

Overall, participants who had accessed support for their mental health reported positive experiences, and this appeared to encourage them to return for support if they needed. They did however acknowledge significant variability in what was available and described how their experience was dependent on the healthcare professional they had contact with.

[After accessing cognitive behavioural therapy via NHS] *"Now that I have almost like a blueprint of services and people to reach out to, then if something similar was to happen or if I was in the position, I'd definitely know where to look, where to go."* 18-25, Black Caribbean, unreported sexuality, London

*"I think my GP was very, very understanding about it. And I've heard, I think, very different opinions from my friends who have had to access kind of clinical help, and there has been different levels of what their doctors have been willing to - or how much they have been willing to engage with them about it. But my GP was very, very understanding and did help me a lot."* 18-25, Mixed White & Asian, bisexual, South East England

### Theme 3.2: Financial means, quality of care and access

Although 8 of the 10 participants who accessed specialist NHS mental health services had positive experiences of their care, there also appeared to be a perception that private therapy services were of a higher quality or that they offered an opportunity to receive support that

was more tailored to their needs. Accessing psychological support through the NHS was sometimes referred to as effectively settling for a second-best option, although not wholly unsatisfactory.

*"All the NHS stuff, obviously, was chosen for me but the private one, I know it's not the right way to pick a therapist, but I actually picked the [...] Well, two reasons, actually. A, the most expensive one I could find 'cause I thought they must then be the best. And the second was the one which had the most letters after their name."* 26-35, Black African, bisexual, East of England

*"I suppose I kind of accept there's going to be waiting times. So I am realistic about what the NHS can achieve and in the timescale that it can achieve it in. It might be – everything could be better but I think it does an okay job and it's never failed me. So it's alright."* 46+, Mixed White & Black Caribbean, gay, London

Many participants explained that access to their preferred modality of psychological support, which for most of them involved talking therapies (including counselling and psychotherapy), was mainly limited by how much it cost. If these therapies were not subsidised by an employer or academic institution, they were generally unable to afford them.

*"I mean, for me, therapy was really kind of the main factor that helped me to start to recover really. I think I would choose a similar route, but if I had money, I would pay for private therapy just so I don't spend nine months on a waiting list. Yeah, I think that's the main kind of barrier really."* 18-25, Mixed White & Asian, bisexual, South East England

### Theme 3.3: Stigma and perceptions of mental health issues

Interviewees explored their thoughts about the role of stigma in shaping service accessibility. This sometimes manifested as internalised feelings of shame, and other times as anticipated stigma from the healthcare worker who they felt would judge them for their sexual practices. Some also felt that they had experiences enacted stigma having been profiled for being gay at the cost of a wider consideration of their health needs.

*"I think that's all the internal self-talk that you have which is all around what society thinks a good boy should or shouldn't do. If you have a different sexual partner every night, people are going to assume that you're a slag. So you walk into the test knowing that these people are influenced by society, even though they are professionals. They may not show it to you because they have been working for a while and they have learned to have a bit of a poker face. But these are internal barriers."* 46+, Mixed White & Black Caribbean, gay, London

*"But then because I was a gay man, the first thing she thought was HIV. And it wasn't [...] There was no internalised homophobia there but I was pissed off that just because a gay man's lost weight, it seemed a bit '90s or '80s. Like, oh my God, you must have HIV or something when actually there could be many other reasons … my doctor didn't even consider that I might have an eating disorder."* 26-35, Black African, bisexual, East of England

Some interviewees described reluctance to seek support for mental health problems, especially through their GP, out of fear that having a mental health diagnosis would be visible on their medical record and could lead to being discriminated against in the future, an issue which may be particularly acute for men already facing multiple forms of marginalisation.

*"I didn't want to get a diagnosis and then if someone asked me on a job application, have you ever seen mental health professional in a professional capacity, and then I would be put in a situation where I think, okay, I'm going to have to lie because I have spoken to them and, yes, diagnosed with depression. And then it destroys my chance for a job that I feel that would have been a really good prospect for me. So I waited a very long time before seeking help."* 46+, Mixed White & Black Caribbean, London

### Theme 3.4: Structural factors, language and geography

This theme explores the structural barriers to using mental health services, including geography, and language issues and immigration status.

Living in an exurban or rural area made finding appropriate services difficult for some participants, who were unsure where they could access care that was tailored to both their physical and mental health needs as MSM. In addition, participants discussed more inconspicuous gay populations outside of cities as limiting their ability to develop a strong LGBTQ+ social network and find support this way.

*"Whether or not they're particularly available in this area, I don't know, having not tried to access them."* 36-45, Mixed White & Black African, gay, South East England

*"Where I live I don't have any friends. I've been living up here for two years now with my stepdad. Yes. No, I don't have any gay friends here."* 18-24, mixed White and Black African, gay, London

In a similar way to building stronger social connections with people they felt akin to, interviewees also described feeling more at ease with a therapist that they shared common demographics with, such as sexual orientation, gender and ethnicity. Some described experiences of alienation from their practitioner, or of being unable to open up completely if they felt that the practitioner would not understand them due to having different lived experiences.

*"I have to say some counsellors I've not been as open to because they've either been female and I've just not felt comfortable to open myself about my life. But I did pay for a counsellor, a gay counsellor, in the past with my sexuality and that sort of stuff. I found that quite helpful."* 46+, Mixed White & Black Caribbean, gay, London

*"There was one therapist who I would go back to and she was the only therapist of colour that I had. And actually there were a lot of things I was honest with her which I couldn't be honest with any of my other therapists 'cause they were all White. So I don't think it is specifically to do with race, like, oh, I need a Black therapist. But maybe it's more to do with I just need a therapist who I can trust 'cause they've been through similar things to me."* 26-35, Black African, bisexual, East of England

One participant also talked about feeling excluded from getting the support he needed due to seeking asylum, and not having an awareness of what was available for him in this situation.

*"I would have liked* [psychological support] *to be fair. I think because I was going through asylum at the time, because I was an asylum seeker I didn't have access to all these things."* 26-35, Black Caribbean, gay, London

This also was true for another participant who felt that language was a barrier to getting the right level of support, in addition to a general sentiment that services did not feel inclusive to him as a Latin American man born outside of the UK.

*"But I don't feel like it's really designed for me. It's designed for everybody. And it's really difficult as a Black migrant to feel anything is designed for me … you see there is no other language on anything or any packet whatsoever. There is no leaders and help saying in other language, even in the NHS website. So it's really not designed for me. It's designed for the general public."* 18-25, Latin American, gay, London

## Discussion

This research provides the first insights into the experiences and mental health needs of Asian, Black and Latin American cis-gender MSM in England and Wales. We explore the perspectives of 29 diverse participants in regards to their own mental health, its importance in their upbringing, their insights into mental health issues among peers and their experiences of using mental health services.

Our analysis shows how the influence of familial culture and upbringing (including experiences of homophobia and the influence of backgrounds with limited openness about mental health) influences perspectives regarding mental health issues and their manifestation. These experiences often negatively impacted health seeking behaviours, which could be compounded by hypermasculine norms valuing self-reliant coping strategies, especially among MSM from Black African and Black Caribbean communities. Norms surrounding masculinity among Asian MSM focused more on success in employment/education as a proxy for resilience, psychological strength and therefore a representation of masculine ideals. These norms constrained participants' willingness and ability to recognise their vulnerabilities. This is in line with research from the USA which found that Black and gay identities were often in conflict with one another and could constrain health seeking behaviour, particularly for mental health, through gendered expectations of what was acceptable [107].

As well as providing a rich support network, facets of the gay scene were also felt to have some negative impacts. The increase in harmful use of alcohol, party drugs and participation in chemsex during periods of poor psychological health was highlighted. Many felt that extracting themselves from these activities was difficult due to how embedded they were in the gay community. For many MSM, these behaviours became habitual at an early point in their adult lives as an actuation of their gay identity; this often persisted through their life course. This complements existing research on how substance use, sometimes intractable from the gay scene, impacts ethnic minority MSM [108–111].

The study also demonstrates systemic barriers and facilitators to accessing mental health services. Access facilitators included previous service use and openness about mental health within social networks. Our analysis compliments existing literature identifying support networks within the gay scene as protective and potentially an opportune space for health promotion, providing an environment to discuss issues among people who feel aligned in their lived experiences [78]. However, caution is warranted as these spaces can sometimes be hostile for MSM from ethnic minority communities [104].

Those living outside of cities who do not have access to visible LGBTQ+ communities have less opportunity to identify sources of support, and for Asian, Black and Latin American MSM, communications from services can feel geared towards White populations leading to cultural and linguistic alienation. Language was identified as a barrier and is known to play a key part in access to healthcare [112] as well as participation in research [113], and future

research should acknowledge participants' first language to mitigate for underrepresentation of people for whom English is not their first language. Migration status should also be considered for its relationship to both mental health outcomes and social inclusion [114,115]. Fear of judgement from healthcare staff, and experiences of being profiled as a homosexually active man at the cost of being consulted more holistically were also concerns that discouraged access. The availability of a diverse healthcare workforce where similitude is expressed through racioethnic or LGBTQ+ relatedness improved the level of comfort and engagement with mental healthcare among participants. These findings highlight the importance of training staff to foster inclusive and culturally competent environments in both primary and secondary healthcare services.

Despite access to free healthcare in the UK, financial means remain a barrier to accessing mental health services. Private services, which could be accessed more quickly than NHS services, were perceived to be of a higher quality and potentially more tailored to specific needs of Asian, Black and Latin American people. This aligns the UK minority ethnic MSM population with other sexual minority populations studied in the US, for whom financial adversity is a barrier to accessing appropriate mental health care [78].

Although the data were collected in 2020, the findings continue to reflect key issues these groups face in accessing mental health support. In addition, in the UK physical and mental health outcomes for ethnic and sexual minorities continue to worsen alongside an ongoing deterioration in mental health service provision in the UK, enhancing the relevance of this research [90,93,116,117]. Further, MSM from ethnic minority backgrounds may face additional stigma negatively impacting mental health following the 2022/2024 global mpox clade IIb outbreak due to intersections between stigma around homosexuality and racism linked to both popular understandings and media depictions of groups most impacted [38,118].

## Strengths and limitations

An important limitation of this study is its context within another qualitative sub-study. As it was one component of a wider enquiry, there may have been less time spent on this section of the topic guide during interviews. As a result, this may have limited the depth of nuance in this research and the differences in respondents' accounts may have been more evident and clearer if there had been more focus on questioning in this area. A further study with a specific interview guide exploring mental health needs would give the space for more detailed exploration of some of the issues described above. Conversely however, as the overarching qualitative study focused primarily on sexual health, sampling bias may have been reduced as recruitment was not on the basis of prior experiences of mental health conditions or service access.

Comparatively few participants with lower educational attainment took part in interviews, and people living with higher levels of social deprivation may not have been included; this group likely has more pronounced concerns relating to mental health and service access which will not be reflected in these findings. It is also important to note that the study does not explore the needs of ethnic minority trans people, who face additional barriers to accessing timely and equitable healthcare [119], and further work should aim to explore their specific experiences.

However, the purposive sampling of non-white ethnic groups resulted in a relatively balanced range of ethnicities. Asian participants formed the largest group in the dataset, a demographic category that encompasses many different cultural and religious backgrounds. Further work would aim to delineate these populations to describe the spectrum of their experiences and identify barriers that correspond to their more specific needs, with a more detailed perspective on ethnic diversity and intersectionality [120–122]. Nevertheless, previous research carried out in the US is consistent with our study's findings, which highlight important unmet

needs among MSM from minority racioethnic backgrounds as a whole despite differences in their cultural and religious practices [15,36,64].

Religious practices were not included in the demographic data, and although some participants discussed a religious upbringing or inclusion within communities of faith, this information was not systematically obtained for all interviewees, limiting the conclusions that could be drawn on the impact of religion specifically on issues around social stigma and self-stigma which are likely to intersect profoundly with ethnic identity [123–126].

## Conclusions

Our findings highlight the importance of multi-system and interdisciplinary interventions to facilitate discussions surrounding mental health within Asian, Black and Latin American MSM communities. In particular, it is critical that the association between poor mental health and transgressing gender norms is challenged. Although relevant for many MSM, for those from these communities these norms appear to pose a greater barrier, possibly because of the greater emphasis on masculinity from their own or their family's cultural or religious values [127–132].

## Supporting information

**S1 File.** Interview topic guide.
(DOCX)

## Author contributions

**Conceptualization:** Andrew Ghobrial, Phil Samba, Alison J Rodger, T Charles Witzel.

**Data curation:** Phil Samba, Emily Jay Nicholls, T Charles Witzel.

**Formal analysis:** Andrew Ghobrial, T Charles Witzel.

**Writing – original draft:** Andrew Ghobrial, T Charles Witzel.

**Writing – review & editing:** Andrew Ghobrial, Phil Samba, Fiona M Burns, Emily Jay Nicholls, Peter Weatherburn, Fiona C Lampe, Isaac Yen-Hao Chu, Alison J Rodger, T Charles Witzel.

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
