## [Decision Letter · Decision Letter 0]

6 Sep 2024

PMEN-D-24-00212

Barriers and facilitators to addressing mental health needs among Asian, Black and Latin American men who have sex with men (MSM) in England and Wales: A qualitative study with participants from the SELPHI trial

PLOS Mental Health

Dear Dr. Ghobrial,

Thank you for submitting your manuscript to PLOS Mental Health. After careful consideration, we feel that it has merit but does not fully meet PLOS Mental Health’s publication criteria as it currently stands. Therefore, we invite you to submit a revised version of the manuscript that addresses the points raised during the review process.

Please ensure that your decision is justified on PLOS Mental Health’s publication criteria  and not, for example, on novelty or perceived impact.

We look forward to receiving your revised manuscript.

Kind regards,

Marc Eric Santos Reyes

Academic Editor

PLOS Mental Health

Journal Requirements:

1. Please send a completed 'Competing Interests' statement, including any COIs declared by your co-authors. If you have no competing interests to declare, please state "The authors have declared that no competing interests exist". Otherwise please declare all competing interests beginning with the statement "I have read the journal's policy and the authors of this manuscript have the following competing interests:"

2. Your current Financial Disclosure states, “Prof Alison J Rodger”. Please indicate by return email the full and correct funding information for your study and confirm the order in which funding contributions should appear. Please be sure to indicate whether the funders played any role in the study design, data collection and analysis, decision to publish, or preparation of the manuscript.

3. In the online submission form, you indicated that "Due to the potentially identifiable nature of the underlying data it will not be made publicly available. The senior author (TCW) will consider individual requests for access."

3. Uploaded as supplementary information.

Additional Editor Comments (if provided):

Reviewers' comments:

Reviewer's Responses to Questions

**Comments to the Author**

1. Does this manuscript meet PLOS Mental Health’s publication criteria ? Is the manuscript technically sound, and do the data support the conclusions? The manuscript must describe methodologically and ethically rigorous research with conclusions that are appropriately drawn based on the data presented.

Reviewer #1: Yes

Reviewer #2: Yes

2. Has the statistical analysis been performed appropriately and rigorously?

Reviewer #1: N/A

Reviewer #2: N/A

3. Have the authors made all data underlying the findings in their manuscript fully available (please refer to the Data Availability Statement at the start of the manuscript PDF file)?

Reviewer #1: No

Reviewer #2: Yes

4. Is the manuscript presented in an intelligible fashion and written in standard English?

Reviewer #1: Yes

Reviewer #2: Yes

5. Review Comments to the Author

Reviewer #1: This is a superb paper which highlights a novel approach to understanding access to mental health services for people with intersectional identities who may be marginalised due to race and sexual orientation. The findings are well contextualised to exisitng literature, presented clearly and discussed with sufficient circumspection. There are only minor typographical/stylistic changes that I can recommend. For example, using a consistent third person narrative, but otherwise the work is highly commended.

Reviewer #2: Please refer to the attached file for some notes for improvement on the specific portions of the article. You have a promising research that is beneficial to the LGBTQ+ community. Please ensure the relevance of the research outcomes to the present time since the data for the study were collected during 2020. My other notes are on the attachment. All the best ahead!

6. PLOS authors have the option to publish the peer review history of their article (what does this mean? ). If published, this will include your full peer review and any attached files.

**Do you want your identity to be public for this peer review?** For information about this choice, including consent withdrawal, please see our Privacy Policy .

Reviewer #1: No

Reviewer #2: No

---

## [Decision Letter · Decision Letter 1]

21 Nov 2024

PMEN-D-24-00212R1

Barriers and facilitators to addressing mental health needs among Asian, Black and Latin American men who have sex with men (MSM) in England and Wales: a qualitative study.

PLOS Mental Health

Dear Dr. Ghobrial,

Thank you for submitting your manuscript to PLOS Mental Health. After careful consideration, we feel that it has merit but does not fully meet PLOS Mental Health’s publication criteria as it currently stands. Therefore, we invite you to submit a revised version of the manuscript that addresses the points raised during the review process.

The manuscript has been re-reviewed by one of the reviewers, and their comments are available below.

The reviewer has only minor requests on the Discussion section remaining.

Could you please revise the manuscript to carefully address the concerns raised?

We look forward to receiving your revised manuscript.

Kind regards,

Helen Howard

Staff Editor

PLOS Mental Health

Journal Requirements:

Additional Editor Comments (if provided):

Reviewers' comments:

Reviewer's Responses to Questions

**Comments to the Author**

1. If the authors have adequately addressed your comments raised in a previous round of review and you feel that this manuscript is now acceptable for publication, you may indicate that here to bypass the “Comments to the Author” section, enter your conflict of interest statement in the “Confidential to Editor” section, and submit your "Accept" recommendation.

Reviewer #2: All comments have been addressed

2. Does this manuscript meet PLOS Mental Health’s publication criteria ? Is the manuscript technically sound, and do the data support the conclusions? The manuscript must describe methodologically and ethically rigorous research with conclusions that are appropriately drawn based on the data presented.

Reviewer #2: Partly

3. Has the statistical analysis been performed appropriately and rigorously?

Reviewer #2: N/A

4. Have the authors made all data underlying the findings in their manuscript fully available (please refer to the Data Availability Statement at the start of the manuscript PDF file)?

Reviewer #2: Yes

5. Is the manuscript presented in an intelligible fashion and written in standard English?

Reviewer #2: Yes

6. Review Comments to the Author

Reviewer #2: Ensure that findings will resonate with the present time.

7. PLOS authors have the option to publish the peer review history of their article (what does this mean? ). If published, this will include your full peer review and any attached files.

**Do you want your identity to be public for this peer review?** For information about this choice, including consent withdrawal, please see our Privacy Policy .

Reviewer #2: No

---

## [Editor Report · Decision Letter 2]

26 Dec 2024

Barriers and facilitators to addressing mental health needs among Asian, Black and Latin American men who have sex with men (MSM) in England and Wales: a qualitative study.

PMEN-D-24-00212R2

Dear Dr Ghobrial,

We are pleased to inform you that your manuscript 'Barriers and facilitators to addressing mental health needs among Asian, Black and Latin American men who have sex with men (MSM) in England and Wales: a qualitative study.' has been provisionally accepted for publication in PLOS Mental Health.

Best regards,

Helen Howard

Staff Editor

PLOS Mental Health